# Evaluation Methods for Citizen Design Science Studies: How Do Planners and Citizens Obtain Relevant Information from Map-Based E-Participation Tools?

Johannes Müller 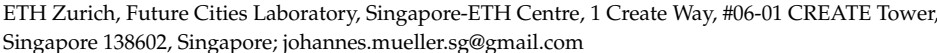

ETH Zurich, Future Cities Laboratory, Singapore-ETH Centre, 1 Create Way, #06-01 CREATE Tower, Singapore 138602, Singapore; johannes.mueller.sg@gmail.com

**Abstract:** A successful e-participation campaign in urban planning relies on good two-way communication between the expert and the citizen. While the presentation of information from planners to citizens is one concern of that topic, we address in this paper the question of how citizens' inputs can be evaluated for map-based e-participation tools. The interest is, on the one side, in the usefulness of the input for the planner and, on the other side, in performing a quick assessment which can provide feedback to the participant via the tool's interface. We use a test dataset that was acquired with an online city planning tool that uses 3D geometries and develop analysis methods from it that can also be generalized for other map-based e-participation tools. These analysis methods are meant to be applied to large datasets and to enhance e-participation methods in urban planning and design to citizen (design) science approaches. The methods range from the calculation of simple parameters and heatmaps over clustering to point pattern analysis. We evaluate the presented approaches by their computation time and their usefulness for the planner and non-expert citizen and investigate their potential to serve as a composite analysis. We found that functions of the point pattern analysis reveal relevant information of the users' inputs but require a simplified presentation. We introduce a spatial dispersion index as an example to present the relations between objects in a clear way.

**Keywords:** public participatory geo-information systems; voluntary geographic information; citizen design science; participatory design; spatial point pattern; spatial dispersion index

## 1. Introduction

The participation of citizens and other stakeholders in the urban planning process has become increasingly important in recent years in order to achieve the United Nations goal of making cities and human settlements inclusive, safe, resilient and sustainable [1]. In the field of e-participation for urban planning, map-based applications are an important solution for a more comprehensive participation approach which is designed not only to provide feedback to the planners but also to educate the participants. The data collected from citizens differ mainly in the way they are collected, and for what purpose they are used. Two main disciplines for applying map-based participation are very prominent. Public participatory geo-information systems (PPGISs) have a long tradition in urban and landscape planning, even before the age of web 2.0 ([2,3]). By contrast, volunteered geographic information (VGI) [4] has developed through the availability of new online data sources and is not necessarily considered only in the context of spatial planning [5]. PPGISs have often been used with the purpose to support the technical knowledge of planners with information about the preferences and attitudes of citizens [6]. On the other side, VGI projects involve larger number of participants who do not necessarily have to fulfil tasks [7].

However, as people become more familiar with map-based technologies, the number of participants in PPGIS studies can easily increase and a sophisticated data evaluation will be unmanageable with the standard tools. Furthermore, the kind of geographic information

becomes more complex. Mueller et al. [8] present the concept of citizen design science, which shifts the perspective from considering "citizens as sensors" for the city [9] towards citizens as non-expert designers and creators of simple city models.

In this paper, we build on this idea and examine how these more complex contributions from citizens can be integrated into the planning process. Our focus is on the assessment of evaluation methods for the geo-data collected with map-based participation tools. In doing so, we concentrate on two aspects: First, we want to identify and develop analysis methods that are suitable for summarizing all submissions. Second, we want to evaluate these methods in terms of their applicability as part of a design dashboard and their usefulness as a composite analysis. The design dashboard is supposed to be presented to a study participant via the tool interface during or after the design process. These quick assessment methods are addressed to non-experts. A method suitable for a composite analysis can be used to summarize submissions from multiple participants and is therefore important for a planner who needs to consider feedback from all participants.

The data that we use to exemplify our methodology are from a pilot dataset that is collected with an online city planning tool that uses 3D geometries. The study site is in Singapore and participants were asked to envision their ideas for a new neighborhood from scratch by using this online tool. They could select 3D objects from a small library that we provided and arrange the objects in the way they would like to build the neighborhood according to their individual preferences.

To address the issue of extracting relevant information for the planning process and for the participants, we will first look at existing data evaluation methods for PPGIS and VGI. We will subsequently describe the kind of data, the tool and the study site. Eventually, we will present the analysis methods and apply them in the results section. The results section includes an assessment of the methods in regards to applicability for a composite analysis and a quick assessment. One of our findings is also the presentation of a new analysis method. Section 6 discusses the outcomes and Section 7 concludes the paper.

## 2. Related Work

Our work is embedded in the overarching topic of participation in urban planning. In this research field, we locate it within the landscape of map-based e-participation tools. As we will present evaluation methods for data collected with these tools, we will also relate them to other existing works in this area.

### 2.1. (E-)Participation in Urban Planning

Sherry Arnstein specified the different levels of citizen participation in the planning process in her ladder of participation [10]. In order to move from the information stage to the empowerment of citizens, it is necessary to establish appropriate communication between experts and citizens. However, this is a frequently observed difficulty [11]. Soudunsaari et al. [12] note that it is difficult for experts to consider citizens' contributions as useful for planning.

The various forms of e-participation as described by [13] offer new solutions for communication between participants and experts by enabling innovative presentation and interpretation of information. Scholars ([14,15]) agree that besides a good information strategy, visual representation of the planning area is a prerequisite for any participatory project. In return, the experts must receive feedback from the participants in an appropriate and organized form [16]. Map-based participation is a popular solution to support this understanding of complex planning relationships.

### 2.2. Map-Based Participation

Map-based participation is an approach to handling citizens' local spatial knowledge. Interesting related work is not only done in this field in the context of participatory planning. It is also considered as citizens' contribution of location-based information. Goodchild [9] characterized these forms of data as voluntary geographic information

(VGI). Prominent examples are OpenStreetMap (OSM), geolocated posts on social media or silently collected user location data (web-scraping methods). These data may also be used for urban planning, but the motivation for the data collection is typically not related to a specific planning site. Datasets and the general nature of VGI are closely linked to the topics of crowdsourcing and citizen science.

Techniques for the evaluation of such data overlap with those for map-based participation in urban planning, which is usually referred to as a PPGIS [2]. McCall et al. [3] elaborate on the differences between a PPGIS and VGI. While VGI allows only a low or medium degree of participation and requires low time investment for participants, a PPGIS requires a high degree of participation and a high time investment. VGI usually does not empower the individual but instead shows the benefits of citizens' contributions on the large scale [7]. A PPGIS on the other hand generates confidence and capacity for the individual participant. Important to the evaluation of these data is the fact that a PPGIS typically contains more in-depth information than VGI data. Trust in the data—one of the five principles of a participatory process according to Verplanke et al. [17]—is established in a PPGIS over time by peer validation and by establishing a transparent process. VGI processes are, however, often not transparent, but its data can usually be validated [3]. A review of more than 200 PPGIS studies ([18,19]) shows that data collection is a main advantage of this participation instrument since data are organized systematically, can be collected on different geographic scales and are usable by various sectors. A drawback is, however, that it is not appropriate for strategic-level questions which require face-to-face discussions.

The data for which we develop analysis methods in this paper are collected in the context of citizen design science studies [8] and, hence, are not clearly assigned to either a PPGIS or VGI. The tool in this study is an online design tool that addresses design challenges for a selected planning site. Participants usually need to familiarize themselves with the tool, the planning area and the task that they must perform, which makes it more of a PPGIS tool. However, it is not limited to a small group of people but is aimed at a larger number of citizens. We want to enhance the process from a one-way to a two-way interactive communication channel, which is also the goal of PPGIS projects. One aim is to establish a quick assessment of the participant's contribution shown on the tool interface, which is a common feature for VGI tools.

### 2.3. Data Evaluation of Map-Based (e-)Participation

Data evaluation has been the focus of research studies for both VGI and PPGIS datasets. The analysis of the data depends on the geometry (point, line and polygon) and its assigned information. Many PPGIS and VGI studies aim to sense features of a location (point) [20]. Examples are wants-maps [21], SoftGIS [19] or grievance reporting [3]. All have in common that participants drop points on a map and provide further information on this location. This can be a binary variable (e.g., like/dislike), a set of categories (e.g., multiple-choice questions) or more complex data (e.g., photos).

Levin et al. [22] present a typical application of data analysis for VGI. The authors compared VGI data sources (OSM, Flickr and Wikipedia) with PPGIS data and visitation counts to identify popularity of places in Victoria, Australia. Their methods ranged from correlations and test statistics to a stepwise multiple linear regression. An example where VGI data were analyzed without reference data is presented by Sun et al. [23]. The authors used geo-tagged Flickr photos to draw conclusions about spatio-temporal behavior patterns of tourists in Vienna. They identified hotspots by applying a Kernel density estimation (KDE). The spatial scan statistic [24], which tests if a one-dimensional point pattern is placed randomly, was used to indicate spatial clusters. These clusters were compared between all four datasets by the number of observations and the spatial dispersion. Guerrero et al. [25] also analyzed geolocated images on social media. Their strategy was to simplify the complexity of the images by first categorizing them and subsequently applying a hotspot analysis and a distance analysis (measured distance to the city center). Mülligann et al. [26]

applied a spatio-semantic analysis to the point features of points of interest in OSM. Their evaluation tools included variograms, point pattern analysis (especially second-order point patterns) and spatial autocorrelation. Many of the techniques mentioned in this paragraph are also applicable to our dataset.

Similarly, PPGIS studies applied several analysis methods to discover spatial relationships. Acedo et al. [27] conducted a map-based survey to detect individuals' geographical sense of place and social capital. The polygons that participants drew during the survey were simplified to points to which the researchers applied point-based (e.g., Ripley's $K$ function), area-based (comparison of frequency distribution between the two datasets and overlap of polygons) and distance-based analyses (distance to the participant's home location). Afterwards, they applied a KDE to overcome the nature of boundaries towards vague geographical areas.

The topic of vernacular geographies is analyzed in a study by Evans and Water [28]. The authors let participants locate places (points) that they perceive as high crime areas. After submission, the participants saw a composite map showing a grid with grey scaled cells according to the relative frequency of points from all other participants. The authors also suggest the use of polygons to indicate the areas which could reveal, similarly, a fuzzy-attributed composite map by overlapping all participants' polygons. Carver et al. [29] built on that idea but replaced, in their tool, the draw function of polygons by a spray. The advantage is that participants could indicate different levels of importance instead of a simple binary response which polygons only allow. To summarize the results, the authors decided to perform both individual analyses as well as create composite maps by overlapping the spray patterns of all participants.

Jankowski et al. [20] present an online study in which participants could sketch polygons on a map and assign attribute information. To summarize the results, the authors counted the polygons of specific attributes and aggregated the assigned responses from all drawn polygons or for sites. For the site-specific preference analysis, they followed the strategy of aggregating cognitive maps [30], which is disaggregation, collective aggregation and individual aggregation. Many of the techniques that we present in this paper are related to individual aggregation, which is considered the best to respect the diversity of responses. Disaggregation refers to the evaluation of individual datasets which are only pooled for comparison. This approach can be used for our dataset to give feedback to the participant by setting it in relation to the other ones.

The abovementioned works make it clear that VGI and PPGIS analyses are used to capture complex information in a spatial context. Several VGI analysis methods focus on comparative studies which are not applicable in our context ([22,31]). PPGIS tools, on the other hand, have only sometimes been used as non-expert design tools [20], so the exploitation of new data formats (such as from our 3D online design tool) remains an open issue. Another gap that we identified is quick assessment of the participants' contributions, which bears the opportunity to provide immediate feedback to each participant. Many authors admitted that some of their analyses required long computation times and were not aimed at a technical implementation in the tool ([25,32]). By providing a quick assessment, the quality of the participatory planning process will be enhanced by educating citizens in planning and GIS ([27,28]). More specific design challenges could also be implemented and hence make tools more playful and interesting for participants ([12,29]).

We contribute to the existing literature by addressing the two mentioned gaps. For new data formats that are collected through map-based e-participation tools, we try to generalize our results so that the data evaluation methods can be further applied independently of the tool. Quick assessment of participants' contributions will strengthen the production of versatile knowledge for both the participants and planners in the existing tool landscape [18]. We will also consider approaches in the context of citizen science. By estimating the computation times and scalability of the methods, we make recommendations for application to large datasets.

The relevance of our work becomes clear with a look at the landscape of map-based e-participation tools. ESRI has recently equipped its ArcGIS Urban tool with an editing function of 3D models [33]. Maptionnaire has added features such as real-time feedback maps and also 3D models (e.g., CityGML and BIM) as additional features [34]. The tool has also been equipped with an Automatic PDF Creator which generates a downloadable summary and a quick assessment of the participant's contribution. Furthermore, urban planning authorities are also gradually opening to creative brainstorming with the public. For example, the Urban Redevelopment Authority (URA) in Singapore held idea competitions for new planning areas [35].

## 3. Tool and Data Description

### 3.1. Tool Description

The online design tool used was a viewer and modifier for 3D objects called the Quick Urban Analysis Kit (qua-kit) [36]. A base map containing predefined 3D objects illustrates the existing spatial context of an exercise. Users may add objects from a menu of predetermined objects (Figure 1a). Editing of the object (e.g., changing the object size and height) is not possible. Therefore, the evaluation will focus on the arrangements of the objects on the map and will not encompass the object's 3D structure. At the current stage of tool development, general design challenges can be addressed by the participants. The presented evaluation methods have the potential to be used for creating further design tasks that include constraints in the design (e.g., the participant needs to build a minimum of residential units). A mock-up for a redesign of the tool user interface is presented by Mueller, Asada and Tomarchio [37]. The reason for using this tool for the study was its accessibility via the browser in combination with the feature to manipulate simple city models. Meanwhile, some tools (e.g., Maptionnaire and ESRI ArcGIS Urban) contain similar features and can be similarly deployed.

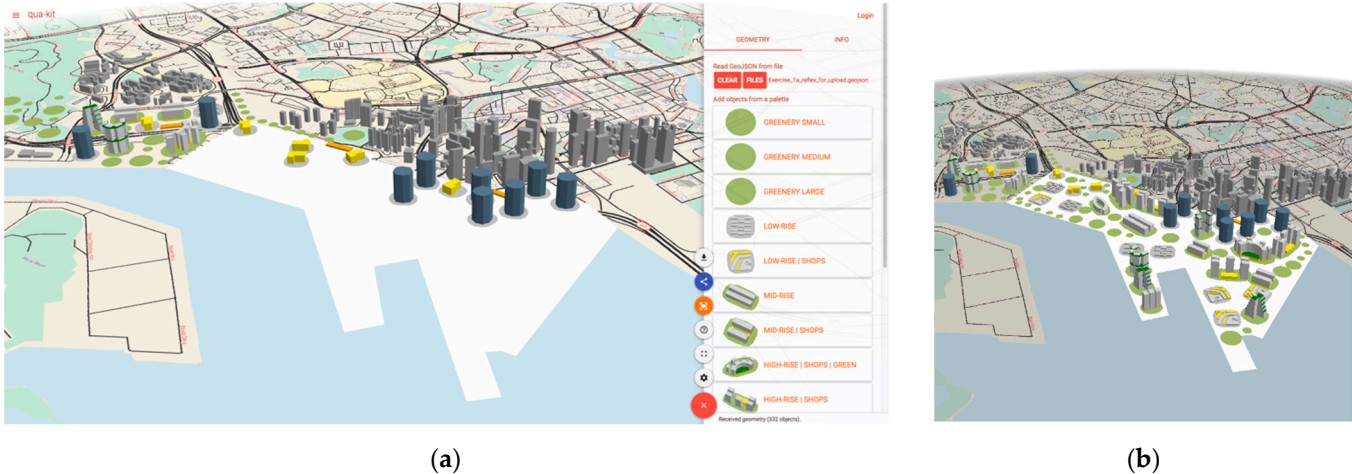

| (a) | (b) |

**Figure 1.** (**a**) Screenshot of the Quick Urban Analysis Kit (qua-kit) user interface. (**b**) Design proposal of one participant which is used as an example for demonstration of evaluation methods of individual submissions in this article.

### 3.2. Study Site

In 2013, the URA in Singapore announced that by 2027, the container terminal in the central area of Tanjong Pagar will be moved west to Tuas, and the resulting 4.5-km$^2$ area would be redeveloped into a high-density mixed-use district. Our research operates within this larger framework [38]. In our discussions with the URA, we were encouraged to develop methods to derive insights from crowdsourced design submissions that planners can use in the design process. As plans have not been fully developed and the site only needs to comply with basic planning regulations, we could test new online participatory planning

methods. Our study entailed the use of the above-described web-based participatory design tool, which was disseminated online through social media, and in public roadshows and workshops, and resulted in over one hundred submitted proposals. Besides, we also used the tool in an experimental setting to ensure the quality of the submitted data. In this paper, we will only look at the submissions from this controlled group (pilot dataset, n = 18) to illustrate our analysis methods.

*3.3. Exercise*

This paper discusses one of the design exercises described by Tomarchio et al. [39]. In this exercise, participants were asked to design a predominantly residential area of around 1 km$^2$. The planning site was marked as a blank white space with no infrastructure and only a few functional buildings (commercial, offices) (Figure 1a). They chose objects from a predetermined library of twelve housing typologies, which are based on existing residential buildings in Singapore, and green spaces in three sizes. The design task for the participants was to envision this space with these given objects. At the time of the study, the evaluation methods presented in this paper have not yet been implemented and tested by the participants, but they could be integrated in a revised version of the tool as described above.

*3.4. Data Analysis*

The core database of qua-kit is geo-referenced data which are stored in the geojson format. A 3D object is—contrary to other architecture software such as CityEngine or Rhino—is represented by its faces, which are polygons. Nonetheless, design criteria such as orthogonality of road connections or block sizes ([40,41]), isovist analysis [42] and symmetry analysis [43] can be calculated but are not applicable to our present pilot dataset. This is because the objects in our exercise represent a class of objects with a certain style and not objects with the exact same typology. As such, an evaluation of the specific building shapes and plots is not suitable for our study. Analyzing the general arrangement of the buildings and objects is, however, useful and, therefore, part of our methods.

All objects have additional information such as the name of the object, the categories they belong to (e.g., low-/mid-/high-rise building, public housing (HDB)/privately developed housing, mixed-use, sky parks, etc.). Next to these qualitative marks, there are also quantitative marks available, such as the number of units one building fits.

In accordance with the identified research gaps, we describe the analysis methods in respect to the usefulness for our two main target groups: the planning expert and the participant. Collective and individual aggregation methods are relevant for the planners, whereas the participant requires quick, individual analysis techniques. For aggregation methods we refer to the complete pilot dataset; for illustrating individual analysis methods, we use the participant's submission shown in Figure 1b.

## 4. Methods

*4.1. Analysis 1: Design Features*

4.1.1. Frequency of Placed Objects

A very simple analysis of the design submission is to calculate the frequency of objects that will be added to the design. A bar chart or pie chart is a common method for presenting the results. The frequency of placed objects (and object groups) can be an indication of the participants' preference for these objects. It is recommended to pre-classify the objects in the library to structure the evaluation into object groups. This simple analysis serves mainly to support the understanding of the rough organization of the participant's design submission. Both the participant and the planner quickly see the percentage of specific objects and object groups.

### 4.1.2. Design Parameters

As described above, we used typologies of existing buildings in Singapore and were thus able to assign the number of units and floors to each building shown the library. As a consequence, we obtained the total number of units, the plot area and the average gross plot ratio of each design submission. With the assumption of an occupation rate per unit, we estimated the number of people residing in that area. As the study site was fixed, it was also possible to determine the population density.

### 4.2. Analysis 2: Heatmaps

### 4.2.1. Qualitative Data: Heatmaps and Kernel Density Estimation

The location of the objects was completely unconsidered in the first analysis method. A heatmap is often taken to indicate the concentration of points. The term commonly refers to an overlay of points or polygons or the above-mentioned KDE. An overlay in our case can be achieved by reducing the 3D geometry object to its plot (ground plan) or by applying a KDE to the centroids of these plots. An aggregation of all participants' submissions was achieved by overlaying each participant's map to a composite map. Differentiation according to the object group or another qualitative mark is strongly recommended, as it would otherwise be difficult to recognize a pattern visually.

### 4.2.2. Quantitative Data: Kernel Density Estimation (KDE)

KDE has the advantage that quantitative data are also included in the analysis. Moreover, it might be better suited for individual feedback as it provides smoother color transitions than heatmaps. The grid size argument of the KDE function from Python's Seaborn package determines the level of smoothness of the map and should not be too small as it increases the computation time. The bandwidth should be selected as a fixed scalar to make all Kernel density estimations comparable with each other and independent of the number of points in each submission. Adding the quantitative mark into the analysis is done by multiplying the object points according to its quantity.

### 4.3. Analysis 3: Clustering

Spatial clustering is another option to determine the spatial distribution of objects. Next to the spatial distribution, any other qualitative and quantitative characteristics can be considered. The purpose of clustering as a data exploration method is to identify the features of points which are assigned to the same cluster. In a spatial context, clusters also indicate hotspots.

### 4.3.1. Non-Hierarchical Clustering (e.g., k-Means Clustering)

We again took the centroid representation of the objects to facilitate the measurement of distances. Depending on the clustering algorithm, only quantitative data can be used for clustering (e.g., k-means). The coordinates of the centroids (and, hence, only the spatial distribution of the points) are the criteria that were chosen to build the clusters in the presented examples.

### 4.3.2. Gaussian Process Clustering

Gaussian process clustering [44] is a machine learning algorithm that takes observed data points as test a dataset to split a space into disjoint groups based on the observed variance function. Every point in space is therefore assigned a probability of belonging to a cluster.

### 4.3.3. Spatial Autocorrelation Statistics

Another algorithm that is considered as clustering [45] is autocorrelation analysis. Moran's I and Geary's C statistics test spatial data for the hypothesis of whether the data are hyperdispersed, clustered or spatially randomly distributed. The spatial data are usually count data with an underlying connected grid, which is required to determine neighbors.

### 4.4. Analysis 4: Point Pattern Analysis

Clustering is a useful method for locating hotspots and quantifying the spatial distribution of objects. However, it is not the most suitable approach to identify and characterize the surroundings of specific objects or object groups. This question has been discussed in theory under the term point pattern analysis. Two steps of the basic point pattern analysis procedure ([46,47]) are also useful for our purpose: determining the data type and selecting appropriate summary statistics. The development of a complete null model is obsolete as there is no interest in generalizing and extrapolating the designs submitted by participants.

In the following, we will again work with the object centroids instead of their plots. The object group and other textual information (e.g., name) of each are considered as (multivariate) qualitative marks (categories), whereas assigned numbers (e.g., number of units) are quantitative marks.

#### 4.4.1. Diversity Indices

The summary statistics for data with qualitative marks consist of simple diversity indices such as the Simpson index [48] or the Shannon index [49] and are also computable for our data. We will use them to indicate the diversity of object groups in each submission. The indices are usually normalized so that they are easily interpretable. They only indicate the general diversity, composition and dissimilarity of the data, without fully considering the spatial relationships between the points.

#### 4.4.2. Common Second-Order Statistics

Other typical summary statistics are distinguished between numerical and functional statistics [47]. The general intensity $\lambda$ indicating the number of points per unit area is a classic example of a numerical statistic. However, functional summary statistics are used more often. Besides the location-based intensity function $\lambda(x)$, there are also second-order statistics that are based on the spatial relationships of point pairs. We will look at the main summary statistics of Python's PointPats package and describe their usefulness and interpretability in our context.

The $G$ function, which is referred to as the $D(r)$ function in [46], is the cumulative distribution function of distances to the nearest neighbor. While this function is a measure for event-to-event distances (meaning the actual centroids of the objects to other centroids), the $F$ function computes point-to-event distances (randomly distributed points to the actual object centroids), indicating the empty space nearest to the event distance distribution ($H_s(r)$ in ibid.). The $J$ function ($g(r)$ function in ibid.) is defined as:

$$J(d) = \frac{1 - G(d)}{1 - F(d)} \tag{1}$$

Compare the Value of the Two Functions

The $K$ function, which is the renowned Ripley's $K$ function [50], is a different popular way to show spatial aggregation, randomness or hyperdispersion. For calculation of the function, there are disks with radius r created around randomly distributed points. The graph of the function shows the number of other event points (our object centroids) within these disks for different r values. Based on Ripley's $K$ function is the normalized $L$ function ($L_2$ function in [46]), which is easier to interpret and is described in detail in the results.

## 5. Results

In this section, we present the results of the methods from Section 4 applied to our dataset. We will also include two discussions: One is on the applicability of the methods to be used as a composite analysis to summarize submissions for the expert, and the other is on the applicability of the methods to be used as a quick assessment and immediate feedback for the non-expert. Based on these discussions, we will introduce a new method (spatial dispersion index for multivariate point patterns) in Section 5.4.3.

### 5.1. Analysis 1: Design Features

5.1.1. Frequency of Placed Objects

The average number of objects which the study participants placed is shown in Figure 2a. Greenery patches are by far the most frequently used objects. This reflects their popularity, but this direct interpretation is only valid to a limited extent as they are used differently than the other 3D objects: They are used to express general greening of areas and overlapping patches are quite common in the submitted designs.

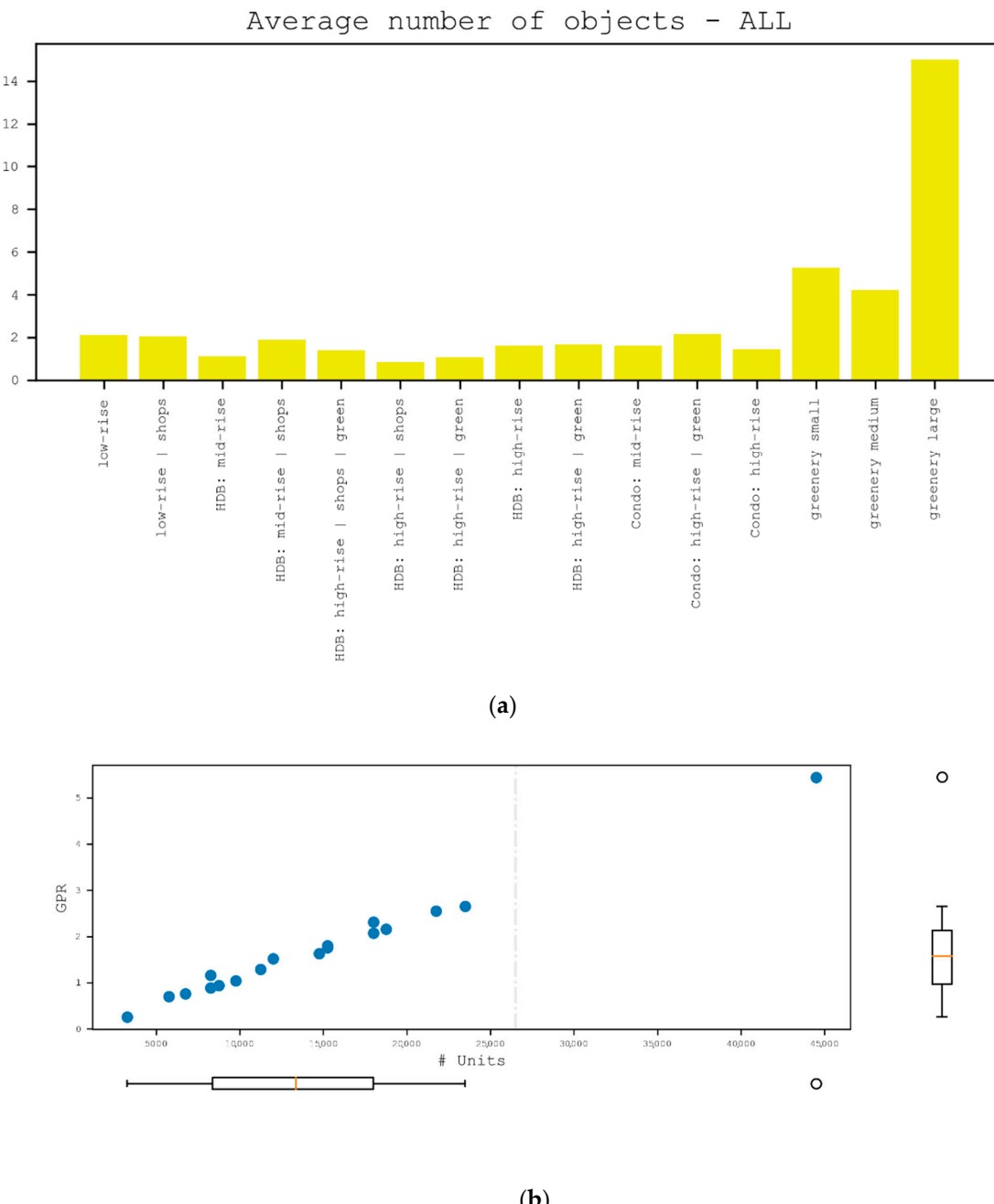

(a)

(b)

**Figure 2.** (**a**) Average number of objects placed by participants. (**b**) Gross plot ratio (GPR) and number of units for each submission. The dashed line indicates the desired number of units by the planning authority.

The analysis is simple to implement and to understand by the participant and is, therefore, a good candidate for a quick assessment. A composite analysis of all participants' submissions is realizable and only biased if there are strong outliers in the number of objects.

### 5.1.2. Design Parameters

The gross plot ratio and the number of units of each submission are shown in Figure 2b. The dashed line marks the number of units the planning authority wishes to build on average in this district. It is apparent that participants placed, on average, less residential objects.

These design parameters are very easy to compute since they are based on basic mathematical operations. An aggregation of all submissions is also possible by simply averaging the values. The design parameters are valuable pieces of information for the participant and the planner. In a design dashboard for the tool user, some parameters need to be explained. One way to present them is to set the individual results in relation to the average of a neighborhood that most people are familiar with, or of the city in total. The design parameters show the strengths of map-based e-participation: qualitative terms (such as density) which are usually expressed in participatory planning processes can be quantified. Additionally, presenting and explaining these design parameters contributes to the design education of the public. Similar to the frequency of objects, the design parameters can also be averaged over all submissions and, therefore, are usable as a composite analysis.

### 5.2. Analysis 2: Heatmaps

### 5.2.1. Qualitative Data: Heatmaps and Kernel Density Estimation

Figure 3a shows the results for the pilot dataset. It is obvious at first glance that many participants placed green spaces (circular green patches) near the water, which can be interpreted as the wish for an accessible waterfront.

The method is easily comprehensible and does not require a long computation time, which is why it can be implemented in a design dashboard for users. However, it makes more sense to use it for a composite map to identify patterns across all submissions. A disadvantage comes with an increasing number of participants; the probability of having a very dispersed pattern increases with each additional participant, meaning that this method is not infinitely scalable. The advantage of this method is that the participants and the planner visually recognize distribution patterns of the design submission.

### 5.2.2. Quantitative Data: Kernel Density Estimation (KDE)

For our study, we used the number of units per building as a quantitative mark. Figure 3b shows the KDE function weighted by the units per building for an individual submission (which is, as mentioned, shown in Figure 1b). It becomes apparent that though the objects are fairly evenly distributed over the area, the distribution of the units is not.

Since the method is similar to the heatmaps with qualitative data, it is computationally efficient and implementable to design dashboards for users. A composite analysis is also applicable and reveals patterns of the distribution of the quantitative marks for all submissions.

### 5.3. Analysis 3: Clustering

### 5.3.1. Non-Hierarchical Clustering (e.g., k-Means Clustering)

The algorithm was applied to a single submission (Figure 3c) but can, theoretically, also be applied to a composite map of all submissions, as independence is no formal requirement for clustering data. However, we strongly advise against this collective aggregation, as the interpretation of results becomes difficult.

Most clustering algorithms work computationally efficiently for a small number of data points. For instance, k-means clustering has a computation time of $O(k^{n \times m + 1})$ if $k$ clusters are computed for $n \times m$ dimensional data points [51]. Therefore, an implementation of this method in a design dashboard is an option.

With this k-means clustering, the participant and the planning expert gain similar knowledge about the hotspots for object groups, as with the heatmap. The clusters may not necessarily represent the most important information for the participant. For the

expert, however, this structuring is helpful to reduce the complexity of the dataset and thus facilitate further analysis.

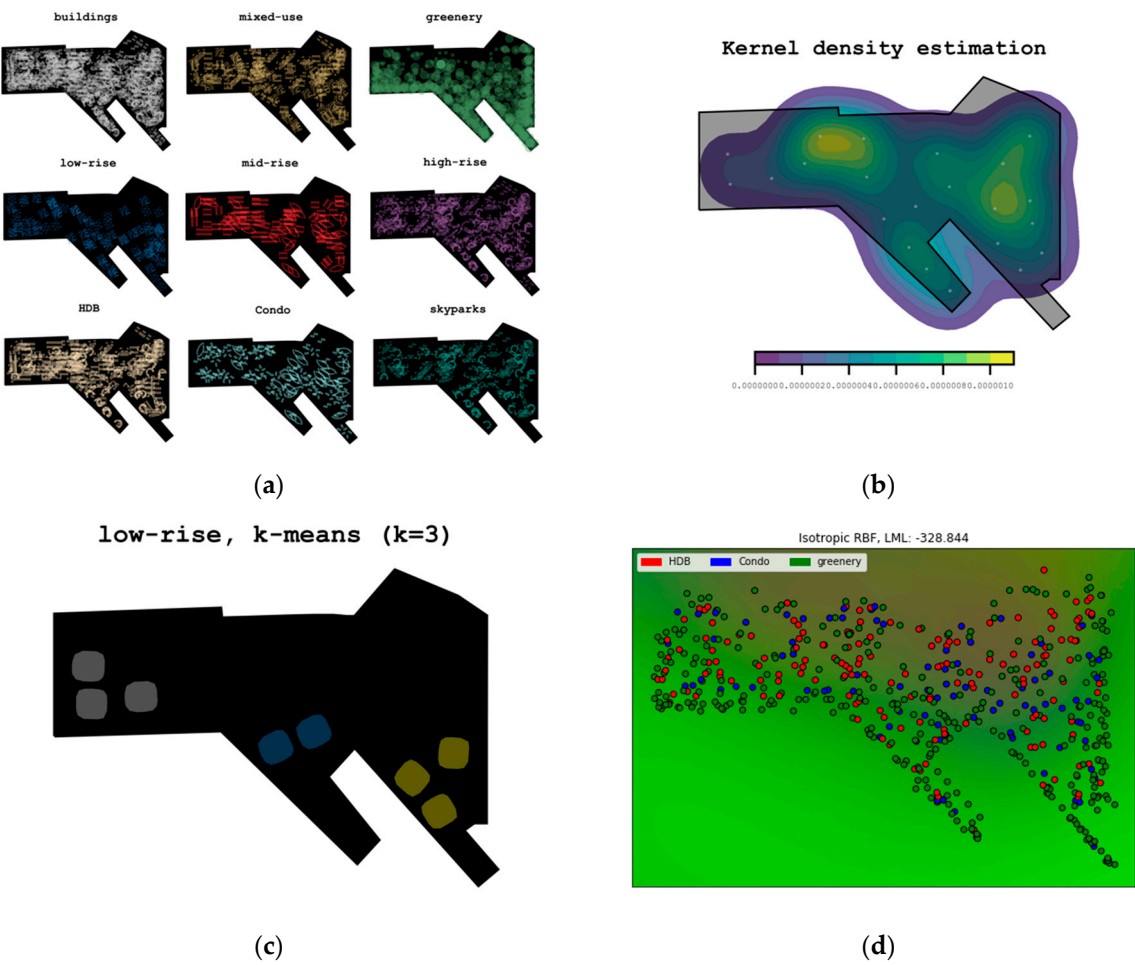

(**a**)        (**b**)

(**c**)        (**d**)

**Figure 3.** (**a**) Heatmap of all submissions separated by object groups. (**b**) Kernel density estimation with number of units as weight for an individual submission. (**c**) Clustering of objects with k-means (here: low-rise buildings) for an individual submission. (**d**) Gaussian process clustering for the object groups "HDB", "Condo" and "greenery".

### 5.3.2. Gaussian Process Clustering

We used a different clustering algorithm for the collective aggregation of data points. Figure 3d shows the application of this clustering method for the different object groups, "HDB", "Condo" and "greenery". The background of the map is colored according to the probability that this area is part of one of these three clusters. The probability of these groups is encoded in the RGB values. An expert or designer gains, from rather clustered data, a quick impression of the areas in which objects were preferably placed. Green areas, for instance, were often located near the waterfront. A serious drawback of this method is the computation time. With an increasing number of points, the calculation cannot be solved efficiently anymore.

### 5.3.3. Spatial Autocorrelation Statistics

Participants normally do not place objects on top of each other, which is why there are no meaningful count data that can be extracted regardless of the grid size. Spatial auto-correlation statistics, hence, do not reveal patterns if applied to an individual submission since the number of points is too small. For a collective aggregation, however, this is a suitable evaluation method. If applied to the object groups, the analysis reveals similar

results as the heatmap method (Section 4.2.1). The algorithm is computationally efficient, but the method is not recommended for implementation in the design dashboard due to the mentioned uselessness for an individual submission.

### 5.4. Analysis 4: Point Pattern Analysis

### 5.4.1. Diversity Indices

We applied the Simpsons index to the number of objects and assigned the object group as a qualitative mark. A value of 1 indicates infinite diversity whereas 0 corresponds to no diversity. For most submissions, the index does not reveal any peculiarity. The submissions generally show an average diversity, and the indices only decrease for submission where one type of object was placed significantly more often than others (e.g., greenery patches). In these cases, the index itself does not reveal which object group was predominantly placed, which we consider as a major weakness of the method. Therefore, we only recommend such indices if used together with a frequency analysis (Section 4.1.1). Such indices are quickly computable and suitable for a composite analysis.

### 5.4.2. Common Second-Order Statistics

Second-order functions were applied for the analysis of individual submissions. Figure 4a shows the *G* function in our example, from which we conclude that the participant placed a building with at least 140 m distance from another building. This pattern is not a complete spatial randomness (CSR) process, which is indicated by the blue line. This peculiarity is caused by the fact that the distances refer to the distance between building centroids and not the actual distances between objects. The F function does not show this behavior because random points are considered instead of event points. The example of the individual submission shown in Figure 4b makes it clear that the participant placed all buildings in neither a clustered nor a dispersed way on the map. The J function has a value of 1 if the underlying process is a CSR process. The fact that the values of larger distances (Figure 4c) tend to increase means that the probability of finding pairs increases in this distance interval (here, between 100 and 240 m).

As the figure is hard to interpret, we look at the provided envelope of this function. Values within the upper and lower bounds (UB and LB) of $\alpha = 0.05$—as seen in our case (Figure 4d)—cannot be considered statistically significantly different from a CSR process. The *L* function normalizes the values and makes them easier to interpret (Figure 4e). *L* (*K*) function values below 1 ($\pi d^2$) correspond to a regular point process (with dispersion), whereas *L* (*K*) function values above 1 ($\pi d^2$) indicate a clustered point process. The similarity with Moran's I is apparent—both methods show the tendencies of point patterns to be clustered. As the functions in Figure 4d,e reveal no statistical significance, we conclude that the participant whose submission is shown in Figure 1b has randomly distributed the objects.

The results of this analysis method are relevant for both participants and experts. A limiting factor, however, is the comprehensiveness of the graphs. While the terms of clustered, random or (hyper)dispersed patterns might still be understandable for most people, the graphs need further explanation. Simulation of the results, which is used for calculation of the envelopes, is generally quite time-consuming, with the result that the use for a quick assessment is limited. Therefore, this evaluation method is only helpful for the expert in the post-analysis to discover statistically significant spatial constellations of objects on the map.

In the following paragraph, we therefore discuss a translation of similar graph results in order to develop more understandable and faster computable outcomes.

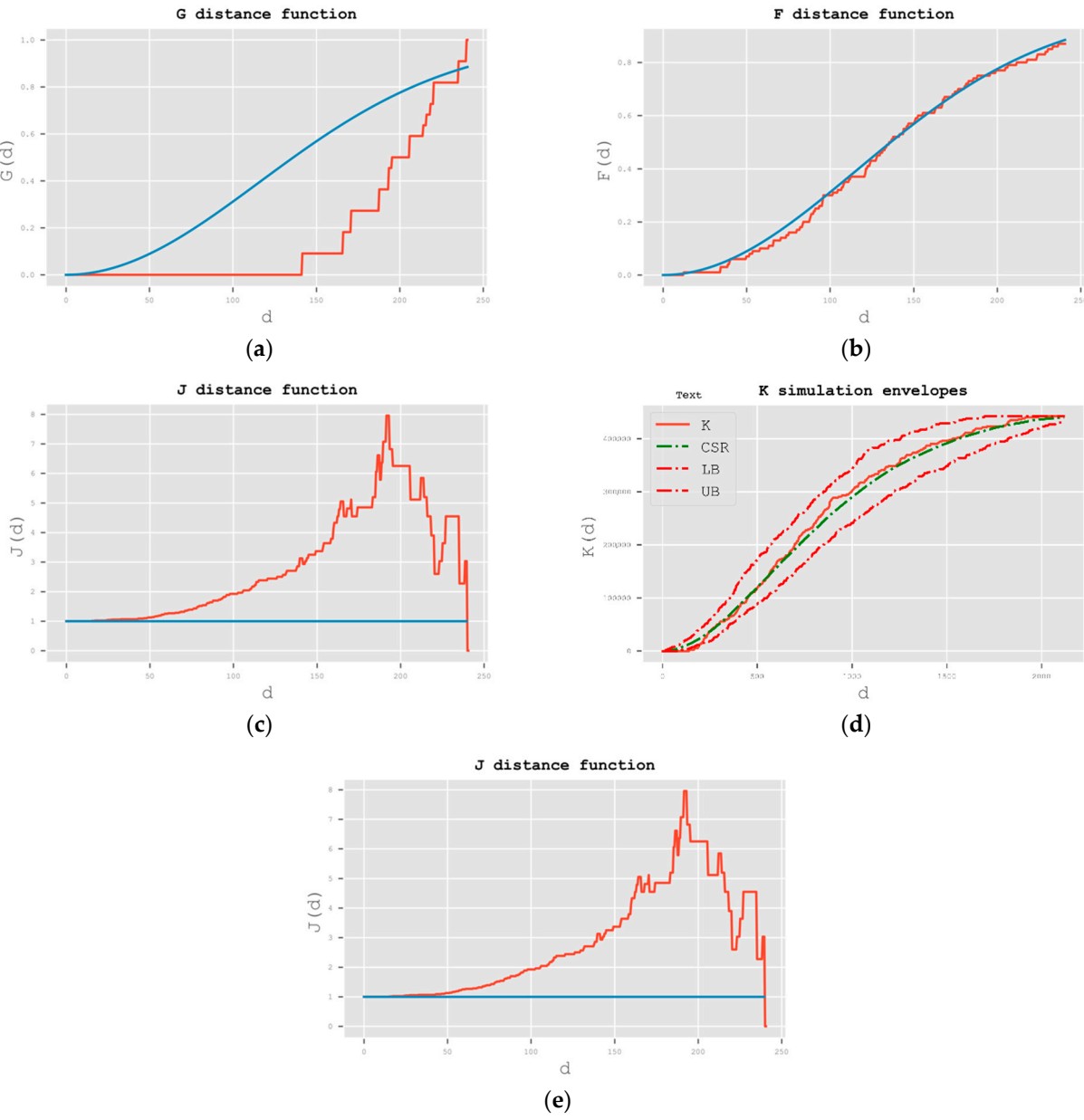

**Figure 4.** (**a**) *G* function, (**b**) *F* function, (**c**) *J* function, (**d**) *K* function (with envelope) and (**e**) *L* function with envelope.

### 5.4.3. Spatial Dispersion Index for Multivariate Point Patterns

Ripley's *K* function is not only useful for univariate but also bivariate or multivariate point patterns. We implemented a variation of this multivariate version by only regarding event-to-event (instead of random point-to-event) distances. As our purpose is not to test the point patterns for null models and extrapolate the data, an event-to-event analysis is more helpful in identifying specific patterns.

For our pilot dataset, we calculated the described event-to-event Ripley's *K* function for object groups. It means that the disks with radius *r* were taken only around the respective points of the object group. The points in each of these disks were counted and normalized by the total number of elements in the object group. The result was a diagram for each object group which contains a graph showing this relative frequency of nearest neighbors for each object group. A major issue of this analysis method is that it requires a lot of computation time as several disks need to be checked (in our case, around 2000). It is, therefore, more efficient to consider the object group membership of the *k* nearest

neighbors (knn) of all objects belonging to this object group instead. The diagrams (Figure 5) provide a similar result and are interpreted in the same way as Ripley's *K* function or the *L* function. If a graph grows rapidly for small *r* or *k*, the participant placed this object group close to the reference object group, which is indicated in the bottom right of each diagram. If the graph only increases for larger *r* or *k*, the object groups are placed apart from each other. Under CSR, the graph would be a linear function with slope 1.

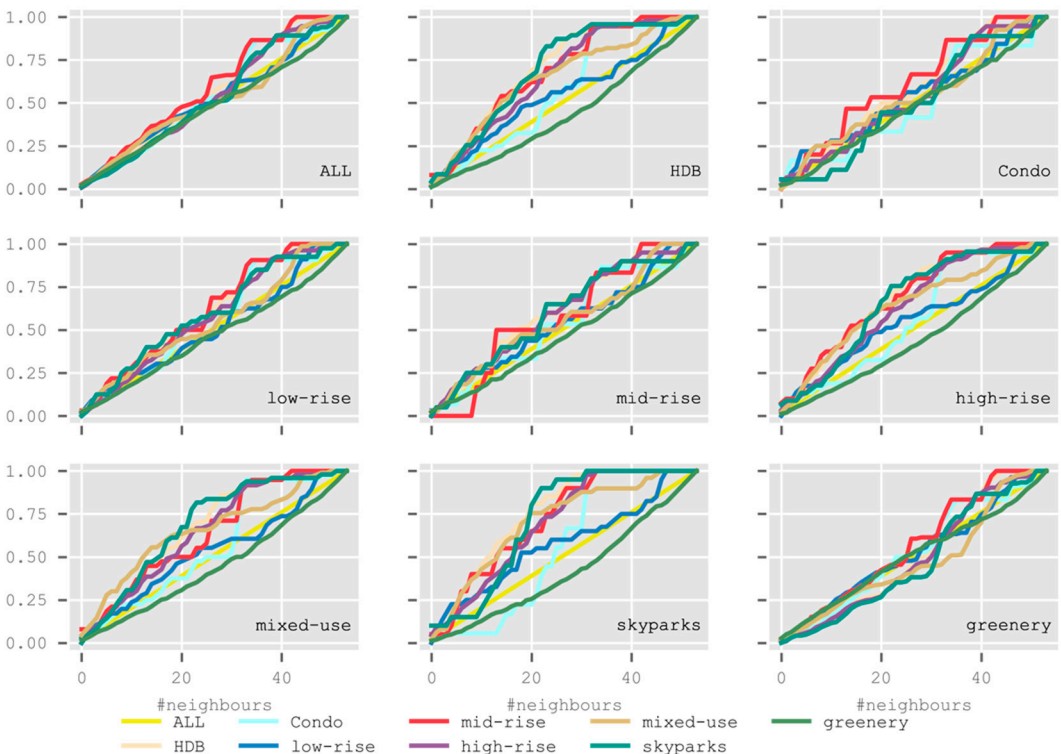

**Figure 5.** Nearest neighbors separated by object groups. The *y*-axis for each diagram shows the cumulated relative frequency (=density function) (CDF) of nearest neighbors, and the *x*-axis shows the nearest neighbor.

As already discussed for the distance functions, the results are not easy to interpret for non-experts. Hence, scholars commonly develop indices to summarize and compare results (e.g., [52,53]).

We propose the construction of an index that is based on the graphs shown in Figure 5. Like the renowned Gini coefficient, we constructed a spatial dispersion index $\kappa(S, T)$ by calculating the area under the graph and compared it to the total area. The idea is shown in Figure 6. The index corresponds to

$$\kappa(T, S) = \frac{B}{A}. \tag{2}$$

where *S* and *T* are object groups. The formal definition is

$$\kappa(S, T) = \sum_{k=1}^{n-1} f(k, S, T) \tag{3}$$

where *S* and *T* are object groups and

$$f(k, S, T) = \frac{\sum_{i=1}^{k} |\{knn(i, S) \in T\}|}{|T|} \tag{4}$$

$$knn(i, S) = \{knn_j(i) \; \forall j \in S\}, \tag{5}$$

where $knn_j(i)$ indicates the $i$-th nearest neighbor of the event point $j$.

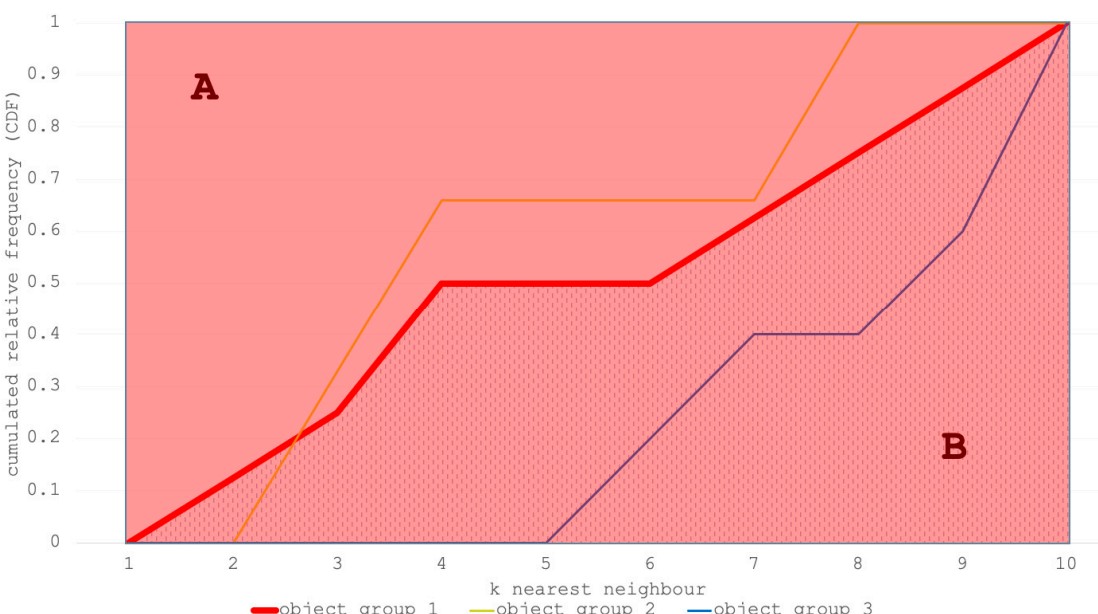

**Figure 6.** The idea of the spatial dispersion index is to consider the area below each graph (**B**, dashed) in relation to the total area (**A**, red). Shown is the example for the area below the red graph (object group 1).

The $f$ function is the formal notation of one graph (object group $T$) in one diagram ($S$ indicates the object group on the right below) and describes the cumulative function of nearest neighbors normalized by the number of elements of object group $T$. It is obvious that $\kappa(S, T) \neq \kappa(T, S)$, which means that the index is not symmetric, and $\kappa(S, T) \in [0, 1]$.

Moreover, under CSR, $\kappa(S, T) = 0.5$, while the occurrence of points of $S$ and $T$ in clusters leads to $\kappa(S, T) > 0.5$ and to $\kappa(S, T) < 0.5$ if pairs of these two object groups are not closely placed together.

This index might still be considered difficult to understand for giving feedback to an individual participant. We therefore suggest converting it into easily understandable text information. For instance, if $\kappa(S, T)$ is above a threshold (e.g., 0.6), the information "The objects $S$ and $T$ are often placed together" could be shown in the user interface.

Table 1 shows the spatial dispersion indices for the participant's submission shown in Figure 1b. It is clearly visible that HDBs and mid-rise buildings are placed near buildings with a sky park. The greenery objects are rather randomly distributed on the map. The same information is concluded from the graphs in Figure 5 (e.g., the greenery graph is close to the line with slope 1 in all diagrams.). However, the index is a much quicker way to communicate the results in a comprehensible way. Another advantage is the opportunity to provide a summary of multiple submissions by taking the mean value of the individual table entries (individual aggregation). For the pilot dataset, we reveal with this new method that, particularly, HDB and buildings with skyparks were placed together. This method quantifies the spatial configuration of objects on maps and, hence, helps both experts and participants to discover spatial patterns without the necessity of a visual representation (in contrast to KDE).

**Table 1.** Spatial dispersion indices $\kappa(S, T)$ for the different object groups of an individual participant. The rows are the originating object groups ($S$), and the columns indicate the object groups of the nearest neighbors ($T$). Red colored cells indicate high index numbers, blue colored ones low indices.

| | HDB | Condo | Low-Rise | Mid-Rise | High-Rise | Mixed-Use | Sky-Parks | Greenery | Buildings | ALL |
|---|---|---|---|---|---|---|---|---|---|---|
| HDB | 0.708 | 0.580 | 0.551 | 0.683 | 0.662 | 0.651 | 0.698 | 0.437 | 0.624 | 0.509 |
| Condo | 0.583 | 0.502 | 0.542 | 0.616 | 0.540 | 0.537 | 0.504 | 0.480 | 0.556 | 0.509 |
| Low-rise | 0.600 | 0.575 | 0.520 | 0.618 | 0.582 | 0.554 | 0.599 | 0.472 | 0.568 | 0.509 |
| Mid-rise | 0.620 | 0.548 | 0.538 | 0.578 | 0.605 | 0.558 | 0.600 | 0.467 | 0.575 | 0.509 |
| High-rise | 0.689 | 0.567 | 0.553 | 0.689 | 0.639 | 0.640 | 0.665 | 0.442 | 0.616 | 0.509 |
| Mixed-use | 0.679 | 0.572 | 0.536 | 0.645 | 0.648 | 0.656 | 0.686 | 0.447 | 0.607 | 0.509 |
| Sky parks | 0.754 | 0.561 | 0.576 | 0.714 | 0.695 | 0.706 | 0.715 | 0.418 | 0.653 | 0.509 |
| Greenery | 0.511 | 0.524 | 0.521 | 0.568 | 0.494 | 0.477 | 0.474 | 0.503 | 0.517 | 0.509 |
| Buildings | 0.643 | 0.567 | 0.539 | 0.646 | 0.611 | 0.594 | 0.627 | 0.457 | 0.591 | 0.509 |
| ALL | 0.562 | 0.541 | 0.528 | 0.598 | 0.539 | 0.522 | 0.533 | 0.485 | 0.546 | 0.509 |

The goal of our study was not only to apply the methods to our dataset but also to generalize their applicability to other similar tools. Therefore, we want to summarize the methods under the aspects of usefulness for planners (by using them for a composite analysis) and for the study participant (by implementing them for quick assessment in the tool interface). These results are listed in Table 2.

**Table 2.** Analysis methods for crowdsourced map-based design.

| | |
|---|---|
| **Analysis 1: Design features** | **1.1. Frequency of placed objects** |
| | *Python package*: collections<br>*Computation time*: Low<br>*Composite analysis possible*: Yes<br>*Usefulness for non-expert and expert*: Revealing the percentage of objects and object categories which can, in some cases, be interpreted as an object's popularity. |
| | **1.2. Design parameters** |
| | *Python package*: geopandas, fiona, shapely<br>*Computation time*: Low<br>*Composite analysis possible*: Yes<br>*Usefulness for non-expert*: Design parameters need to be presented with a short explanation which indirectly supports education of the study participants; comparison of the parameters to existing districts helps to locate own design proposal (e.g., in terms of density).<br>*Usefulness for expert*: Extracting design indicators from non-experts' proposals. |
| **Analysis 2: Heatmaps** | **2.1. Qualitative data: Heatmaps and Kernel density estimation** |
| | *Python package*: geopandas, fiona, shapely<br>*Computation time*: Low<br>*Composite analysis possible*: Yes<br>*Usefulness for non-expert/expert*: Quick visual assessment of spatial distribution of objects and object groups. |
| | **2.2. Quantitative data: Kernel density estimation (KDE)** |
| | *Python package*: Seaborn.kdeplot<br>*Computation time*: Low<br>*Composite analysis possible*: Yes<br>*Usefulness for non-expert/expert*: Quick visual assessment of spatial distribution of quantitative data (e.g., number of units). |

**Table 2.** *Cont.*

| | |
|---|---|
| | **3.1. Non-hierarchical clustering** |
| | *Python package*: Sklearn.cluster, pysal<br>*Computation time*: Low<br>*Composite analysis possible*: Yes, but not advisable<br>*Usefulness for non-expert*: No, heatmaps are the more intuitive alternative.<br>*Usefulness for expert*: Clustering reveals more insightful patterns than heatmaps or KDE. |
| **Analysis 3: Clustering** | **3.2. Gaussian process clustering** |
| | *Python package*: Sklearn.gaussian_process<br>*Computation time*: High<br>*Composite analysis possible*: Yes<br>*Usefulness for non-expert*: No, because the method requires some explanations; though the output can be visualized, it is not applicable for a quick assessment due to the high computation time.<br>*Usefulness for expert*: Planners need to be familiar with the interpretation of the visual output, which is similar to heatmaps. |
| | **3.3. Spatial autocorrelation statistics** |
| | *Python package*: pysal<br>*Computation time*: Low<br>*Composite analysis possible*: Yes<br>*Usefulness for non-expert*: No, as the method only works for count data, and object counts are commonly too small for individual submissions.<br>*Usefulness for expert*: The method works best when being applied as a composite analysis; it reveals an overall preference for locations of objects and object groups. |
| | **4.1. Diversity indices** |
| | *Python package*: pointpats<br>*Computation time*: Low<br>*Composite analysis possible*: Yes, but not advisable<br>*Usefulness for non-expert/expert*: The common diversity indices need explanation; they indicate the diversity of the appearance of objects but do not exploit information of their spatial distribution. |
| **Analysis 4: Point Pattern Analysis** | **4.2. Common second-order statistics** |
| | *Python package*: pointpats<br>*Computation time*: Medium<br>*Composite analysis possible*: No<br>*Usefulness for non-expert*: No, as the method would require too much explanation.<br>*Usefulness for expert*: The method quantifies the spatial relation of objects and object groups towards each other. |
| | **4.3. Spatial dispersion index for multivariate point patterns** |
| | *Python package*: pointpats<br>*Computation time*: Low<br>*Composite analysis possible*: Yes, but only for the indices, not for the graphs.<br>*Usefulness for non-expert/expert*: The method requires a short introduction to the interpretation of the indices; the knowledge revealed is similar to that from the common second-order statistics. |

## 6. Discussion

In this section, we want to point out the significance of the work presented before discussing its limitations and open research questions.

The evaluation methods presented in the literature review (especially those for VGI and PPGIS studies in Section 2.3) aimed to answer research questions from scholars for specific case studies. The aim of most literature is not to discuss these methods in detail but to use them to prove or disprove research hypotheses and find relationships in the data.

This article, in contrast, aims to present a set of methods that are generally applicable to datasets collected with map-based e-participation tools. We want to provide an overview

of evaluation methods so that they can be implemented in participatory GIS tools and automate their data analysis process. Most evaluation methods in the literature have not been assessed for their applicability for generalization to other tools. Our results, as listed in Table 2, will be applicable to many other (participatory) 3D and GIS tools as they work with point representation of objects, which is the simplest form of visual representation. Implementing these methods for tools such as Maptionnaire or ArcGIS Urban will only require minor adaptations, as their data representation of the visual city model is slightly different from the qua-kit tool we used for our study.

We evaluated the methods in terms of their usability for a composite analysis or a quick assessment. This assessment provides both researchers and tool developers with an overview of suitable evaluation methods that are relevant for the two most important stakeholders: the citizens involved and the planning expert. Both of these groups will immediately benefit from an implementation. The citizens will receive concise, insightful feedback to their submissions, and the planners will be able to automatize the summary of relevant design parameters from the public.

The first critical point is that a visual representation of the space is crucial to obtain correct information from the participants. In large online participation studies, this can never be fully guaranteed, and it will always remain an assumption that people perceive the represented space as it really is. One solution is to address this problem with a more precise visual representation of the objects or entire virtual 3D realities, which can even go as far as photorealistic representations [54]. Another solution is to support additional information material (e.g., videos, texts and interactive elements [55]), which not only supports the formation of opinions but also a feeling for the environment of the planning site.

Another argument against massive online participation campaigns is that they do not really establish a connection between planners and citizens. Although the methods presented provide a range of techniques to provide automated feedback to the participant, critics may express concern that this is not a real feedback loop. An automatically generated response to the participant's submitted geographic data via the tool interface does not establish full communication. According to [56], our approach would be categorized as a limited two-way consultation. For this reason, digital planning tools need to be further developed into collaborative platforms where map-based e-participation is one of the communication channels. An interesting question for future research is how these human aspects are already represented in the design. If multiple communication channels exist, scholars can have participants describe their design proposal and then apply a semantic analysis to compare such data with spatial information analysis [32].

Another question is how new online participatory tools should be designed. The one discussed in this article allows flexible movement of objects. It is worth thinking about restricting this freedom and allowing a more parametric design approach for user submissions [57].

## 7. Conclusions

In this paper, we discussed evaluation methods for map-based participatory online tools. Recent developments in the landscape of tools have enhanced the quality and kind of data that planners can collect in participatory studies. The tool with which the presented pilot dataset was collected resembles more of a non-expert design tool than a map survey tool. It is used in the context of a citizen design science study which has characteristics of both PPGIS and VGI studies.

The presented analysis methods capture design-relevant features that can be used in further phases of the design process. In this article, we present several methods for the evaluation of geodata collected with map-based e-participation tools. We apply them to a small pilot dataset and evaluate them in terms of applicability and relevance for planners and participants. For our study site in Singapore, we use these methods to show that the participants in their models have built less densely populated districts than proposed by

the planning authority and prefer green spaces near the waterfront. The public housing buildings were usually placed close to condominiums.

A decisive criterion for the usefulness of a method for planners is whether it is suitable for combining multiple submissions. An evaluation method that is relevant to the citizen must be easy to understand or explain and require a short computing time to be implemented in the interface of the tool. By introducing the spatial dispersion index, we demonstrate that results from efficient algorithms such as k nearest neighbors can be transformed into easily interpretable indicators. Furthermore, design parameters and heatmaps or kernel density estimators are evaluation methods that are well suited for implementation in participation tools to provide automated feedback to citizens. For planning experts, heatmaps, clustering approaches and average design parameters offer a good solution that can serve as a composite analysis for all submitted data.

Application of the presented methods is not limited to the presented data and tool. Most of the methods can be used for different design scales and for other types of geometry, as they can be applied to point geometries with associated qualitative or quantitative data.

**Funding:** The research was conducted at the Future Cities Laboratory at the Singapore-ETH Centre, which was established collaboratively between ETH Zurich and Singapore's National Research Foundation (FI 370074016) under its Campus for Research Excellence and Technological Enterprise programme.

**Institutional Review Board Statement:** The study was conducted according to the guidelines of the Declaration of Helsinki, and approved by the Ethics Committee of ETH Zurich (EK 2018-N-33, 05/07/2018).

**Informed Consent Statement:** Informed consent was obtained from all subjects involved in the study.

**Data Availability Statement:** The data presented in this study are available on request from the corresponding author. The data are not publicly available due to missing permanent resources for servers.

**Acknowledgments:** The author wants to thank his colleagues Artem Chirkin, Hangxin Lu, Katja Knecht, Jonathan Woenardi, Ludovica Tomarchia and Pieter Herthogs for their support. The author would like to thank in particular Gerhard Schmitt for his support and supervision of the study.

**Conflicts of Interest:** The authors declare no conflict of interest.

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
