# Peer review of "Evaluation Methods for Citizen Design Science Studies: How Do Planners and Citizens Obtain Relevant Information from Map-Based E-Participation Tools?"

_ijgi, doi:10.3390/ijgi10020048_

Round 1

Reviewer 1 Report

Dear Author,

It is an interesting paper to read, unfortunately, I am a little bit distracted by the text-align issue.

Line 47-54: Can you help me identify or point out what is actually the aim of this paper? You using two "on the other hand". I find it difficult to grasp the aim of this research from reading this paragraph.

Figure 1: I would suggest that you modify this figure to help the reader understand the workflow of this research.

Section 2 related work: The content of each sub-section is not balanced. Subsection 3 (line 120-205) is too long and should be rearranged to make it clear and concise.

Section 6. Discussion: I would suggest that you add more content to this section, in specific the discussion of Table 2, and how your results can be used or adapted by other researchers in the future.

Finally, please check again how to write the references. (An example to be check, see Reference [1]). Is it U. Nation = United Nation?

Reviewer 2 Report

This is an interesting piece of work which has a great potential to be used in real-world applications. The author’s approach to public engagement in urban planning and urban design is creative and worth pursuing. I suggest that the paper as it stands can be accepted after addressing the following comments:

#1 The title “Evaluation methods for citizen design science studies” neither describes what your paper is about nor gives a clear hint as to what your work’s focus is. If possible, it’s better to find a title that highlights the empirical significance of your work.

#2: Figure 1 does not add any value. It is not common to have figures to outline the structure of your manuscript. Please remove it and provide a concise description of these sections in the preceding paragraph.

#3: Section 2.1 “(E-) participation in urban planning” is not directly related to your work. Could be removed.

#4: Discussion Section: The implications of your findings are missing. Please add two paragraphs 1) to relate your findings to the previous research and existing tools/methods and elaborate on how your findings contribute to knowledge in the field and most importantly what distinguishes your  tools/methods from other tools (e.g. ArcGIS Urban, Maptionnaire, etc.). 2) What does your research mean for the public, in general, and urban planning & design community, in particular. In other words, what is the impact of your findings to a larger context.

I wish the author best of luck in their important and impactful work.

Reviewer 3 Report

The introduction clarifies the purpose of this study and supports the significance of this study with strong arguments. The selection of the case study area is a very intriguing choice, and one of the strongest points of the paper. However, the present structure of the paper makes it difficult to read. For instance, Figure 1 can be put at the beginning of Section 3.

The discussion could be more in-depth. I suggest the authors include some information related to the methods for map-based design. The detailed description of different GIS procedures might be reduced.  such as Table 2 from the conclusion.  Also, the discussion can include some comparisons with other studies (i.e., the finding of map-based design and map-based e-participation), as this is the innovative aspect of the article.

Line 527, Line 529, Line 531, and Line 532: Equations number should be added.

The style of references does not meet the journal requirements.

Round 2

Reviewer 3 Report

Accept.